# Necessary Condition of Self-Organisation in Nonextensive Open Systems

**DOI:** 10.3390/e25030517

**Published:** 2023-03-17

**Authors:** Ozgur Afsar, Ugur Tirnakli

**Affiliations:** 1Department of Physics, Faculty of Science, Ege University, Izmir 35100, Turkey; 2Department of Physics, Faculty of Arts and Sciences, Izmir University of Economics, Izmir 35330, Turkey

**Keywords:** S-theorem, q-renormalized entropy, complexity measures, logistic map

## Abstract

In this paper, we focus on evolution from an equilibrium state in a power law form by means of *q*-exponentials to an arbitrary one. Introducing new *q*-Gibbsian equalities as the necessary condition of self-organization in nonextensive open systems, we theoretically show how to derive the connections between *q*-renormalized entropies ($\Delta {\tilde{S}}_{q}$) and *q*-relative entropies ($KL_{q}$) in both Bregman and Csiszar forms after we clearly explain the connection between renormalized entropy by Klimantovich and relative entropy by Kullback-Leibler without using any predefined effective Hamiltonian. This function, in our treatment, spontaneously comes directly from the calculations. We also explain the difference between using ordinary and normalized *q*-expectations in mean energy calculations of the states. To verify the results numerically, we use a toy model of complexity, namely the logistic map defined as $X_{t+1}=1-aX_{t}^{2}$, where a∈[0,2] is the map parameter. We measure the level of self-organization using two distinct forms of the *q*-renormalized entropy through period doublings and chaotic band mergings of the map as the number of periods/chaotic-bands increase/decrease. We associate the behaviour of the *q*-renormalized entropies with the emergence/disappearance of complex structures in the phase space as the control parameter of the map changes. Similar to Shiner-Davison-Landsberg (SDL) complexity, we categorize the tendencies of the *q*-renormalized entropies for the evaluation of the map for the whole control parameter space. Moreover, we show that any evolution between two states possesses a unique $q=q^{*}$ value (not a range for *q* values) for which the *q*-Gibbsian equalities hold and the values are the same for the Bregmann and Csiszar forms. Interestingly, if the evolution is from a=0 to $a=a_{c}≃1.4011$, this unique $q^{*}$ value is found to be $q^{*}≃0.2445$, which is the same value of $q_{sensitivity}$ given in the literature.

## 1. Introduction

The main problem for researchers who are interested in entropy-based measures is discerning which measure would be the most suitable one for the complex system under consideration. The definition of ‘suitable’ implies the measure which represents a behaviour that is compatible with the dynamics of the system among definitions of the measures in the literature. For example, it has been expected that the measure is able to make a distinction among possible phases of the system as the parameter set of the system slightly changes. Despite the numerous definitions [1,2,3,4,5,6,7,8], the measures can be categorized into three types (Figure 1), namely type-I, type-II and type-III that are similar to SDL complexity, formally [5]. The first type considers measure as a monotonically increasing function of disorder. In the second type, measure is a convex function of disorder. Hence, it is a minimum for both completely order and completely disorder, and a maximum at a point between them. In the last type, measure is a monotonically decreasing function of disorder. The crucial point in this classification is that classic notion of entropy by Shannon [9] is associated with the degree of disorder. It should be noted that the control parameter of the system can also be used as the degree of disorder if the Shannon entropy *S* is a monotonically increasing function of some system parameter, say *a*, as the system evolves in its parameter space from *a* to a+Δa [10].

Even if it seems that changing the parameter in space is not directly related to time, it always takes time from one state to another for a real system since an evolution depends on the change of conditions, i.e., of parameters, with time [11]. For the evolution of dissipative dynamical systems which pave their way to successive stable branches, and then to successive chaotic band mergings as the parameter set of the system slightly changes, the classic notion of the entropy by Shannon, the relative entropy by Kullback-Leibler and the renormalized entropy by Klimontovich are good examples for entropy-based measures of type-I, type-II and type-III, respectively [10]. The Shannon entropy monotonically increases from the first branch to the most chaotic state (type-I). The relative entropy increases monotonically from the first branch up to the edge of chaos, and then decreases monotonically up to the first chaotic band merge (type-II). The renormalized entropy decreases monotonically from the first branch up to the edge of chaos (type-III), and then increases monotonically up to the first chaotic band merge (type-I). In other words, when the sequence of branches emerges, the relative order increases, i.e., the measure of complexity, the renormalized entropy decreases [12].

The behaviour of the renormalized entropy indicates the relative degree of order in the system as first suggested by Haken [13] in the context of self-organization. The S-theorem that is a basis for the method of renormalized entropy was proved for the transition from laminar to turbulent flow [14]. Such a kind transformation confirmed that turbulent structures are more ordered, that implies highly organized, than laminar [15]. Moreover, Rayleigh–Benard convection [16,17], Taylor instability experiment [18], bacterial [19] and Dictyostelium discoideum [20] colonies are some typical examples in which the most ordered spatial patterns emerge in the phase spaces via changing conditions of the systems, that indicates a high level of self-organization.

The Shannon entropy expression, setting $k_{B}=1$, reads
(1)$$S=-\sum_{i}p_{i}\ln p_{i}$$
where $p_{i}$ are the probability of an event *i* of a sample set. Maximizing the Shannon entropy of the system subject to suitable constraints (namely, mean energy and probability normalization constraints), using Lagrange multipliers method, one can obtain the equilibrium distribution as
(2)$$p_{i}^{eq}=\frac{e^{-\beta \epsilon_{i}}}{Z}$$
where β is the inverse temperature and $Z=\sum_{i}e^{-\beta \epsilon_{i}}$ is the partition function. For the evolution to the equilibrium state from an arbitrary one ($p\to p^{eq}$), the Shannon entropy can be used to find the difference between the entropies ${\Delta S}^{\left(p\to p^{eq}\right)}=S\left(p^{eq}\right)-S\left(p\right)\geq 0$ that is known as the second law of thermodynamics. However, from both Boltzmann’s H-theorem and Gibbs theorem, it is well known that this inequality is only valid for an isolated evolution between the states. Hence, it is violated for the evolution of open systems that exchange of energy/matter with its surroundings is allowed. In other words, these theorems state that ${\Delta S}^{\left(p\to p^{eq}\right)}$ equals relative entropy with a limitation that the mean energy 〈E〉 remains constant (i.e, ${\langle E\rangle }^{p^{eq}}={\langle E\rangle }^{p}$) [21]. Therefore, in a process of such an evolution (from *p* to $p^{eq}$) as long as mean energy is the same, from these theorems, it follows that
(3)$$\Delta S=S_{eq}-S=\sum_{i}p_{i}\ln\frac{p_{i}}{p_{i}^{eq}}=K(p||p^{eq})\geq 0$$
where $p^{eq}$ and *p* are probability distributions corresponding to the equilibrium state and arbitrary one, respectively. Equation (Equation 3) implies that the equilibrium state has the greatest disorder (or chaoticity) as compared to the arbitrary state. The usual expression of the relative entropy $K\left(p||r\right)=\sum_{i}p_{i}\ln\frac{p_{i}}{r_{i}}$ gives the entropy produced by the change from the state *p* to the state *r*. Whenever $r=p^{eq}$, it can be written in terms of free energy differences of the states
(4)$$K(p||p^{eq})\propto \left(F-F^{eq}\right)$$
where $F={\langle E\rangle }^{p}-\frac{1}{\beta }S$ and $F^{eq}={\langle E\rangle }^{p^{eq}}-\frac{1}{\beta }S_{eq}$.

Due to the strong limitations on the second law, Klimontovich introduced his S-theorem, where ‘S’ stands for ‘self-organization’, that makes it possible to analyze open systems in terms of the Gibbs theorem [21]. According to the S-theorem, it is possible to compare distinct states, which are the equilibrium and a non-equilibrium stationary state, under dissipation of the energy that implies the evolution of an open system. To compensate the dissipation, a new mean energy equality ${\langle E\rangle }^{{\tilde{p}}^{eq}}={\langle E\rangle }^{p}$ similar to the Gibbs’ equality via a renormalization procedure $p^{eq}\to {\tilde{p}}^{eq}$ is defined. After such a renormalization, the renormalized entropy $\Delta \tilde{S}$ is defined as (noticing that evolution is from ${\tilde{p}}^{eq}$ to *p*)
(5)$$\Delta \tilde{S}={S-\tilde{S}}_{eq}=-\sum_{i}p_{i}\ln\frac{p_{i}}{{\tilde{p}}_{i}^{eq}}=-K(p||{\tilde{p}}^{eq})\leq 0$$
where ${\tilde{p}}^{eq}$ and *p* are the renormalized equilibrium and non-equilibrium stationary states, respectively (all proofs regarding Equations (Equation 3)–(Equation 5) will be given in the next part of the paper).

In Section 2, we show that the definition of mean energy equality (and also Equation (Equation 5)) in the concept of the renormalized entropy by Klimontovich is valid for an evolution to/from the canonical equilibrium distribution of exponential form in Equation (Equation 2). We also discuss the connections among the Shannon, the Kullback-Leibler relative and the renormalized entropies within a thermodynamic perspective.

In Section 3, we theoretically show how to apply this procedure on a nonextensive open system, whose generalized canonical probability distribution is of *q*-exponential form
(6)$$e_{q}^{x}={\left[1+(1-q)x\right]}^{1/(1-q)}\;\;.$$

This distribution is the one that comes from the maximization of the Tsallis entropy given by [22]
(7)$$S_{q}=\frac{1-\sum_{i}{p_{i}}^{q}}{1-q}\;,$$
and it recovers the Shannon entropy $S=-\sum_{i}p_{i}\ln p_{i}$ in the limit q→1 as a special case. It is well known that the maximization of the Tsallis entropy subject to the ordinary constraints ($\sum_{i}p_{i}\epsilon_{i}=\langle E\rangle$ and $\sum_{i}p_{i}=1$) yields a canonical distribution in a *q*-exponential form,
(8)$$p_{i}^{ord}=\frac{{e_{2-q}}^{-\beta^{*}\epsilon_{i}}}{Z(\beta^{*})}\;,$$
where $1/Z(\beta^{*})$ is a normalization constant [23]. On the other hand, normalized *q*-expectation is employed instead of the ordinary constraints, ($\sum_{i}\left[p_{i}^{q}\epsilon_{i}/\sum_{i}p_{i}^{q}\right]={\langle E\rangle }_{q}$ and $\sum_{i}\left[p_{i}^{q}/\sum_{i}p_{i}^{q}\right]=1$), the canonical probability distributions reads
(9)$$p_{i}^{\left(nor\right)}=\frac{{e_{q}}^{-\hat{\beta }\left(\epsilon_{i}-{\langle E\rangle }_{q}\right)}}{Z(\hat{\beta })}$$
where $\hat{\beta }=\beta /C_{q}$ and $C_{q}=\sum_{i}{\left(p_{i}^{nor}\right)}^{q}$ [22]. Respectively, when the generalized version of canonical distributions in Equations (Equation 8) and (Equation 9) put into two kinds of *q*-generalized relative entropies, which are well known Bregman $K_{q}^{B}\left(p|\right|p^{ord})$ and Csiszar types $K_{q}^{C}\left(p|\right|p^{nor})$, it can be shown that the generalized relative entropies are associated with the *q*-generalized version of free energy differences of the states [24]:
(10a)$$\begin{array}{c} K_{q}^{B}\left(p|\right|p^{ord})\propto \left(F_{q}-F_{q}^{ord}\right) \end{array}$$
(10b)$$K_{q}^{C}\left(p|\right|p^{nor})\propto \left(F_{q}-F_{q}^{nor}\right)$$
that are similar to the Equation (Equation 4). It can be also noted that there is one more version of the formalism using unnormalized *q*-expectations in the constraints. However, it is shown in [25] that all these versions are equivalent to each other.

In the following subsections, we explain the necessary condition (on the *q*-mean energy equality) for self organization of open systems using the Bregman and Csiszar forms of generalized relative entropies, respectively. Klimontovich himself as well as some recent efforts on the generalization of the renormalized entropy [26,27] have invoked a predefined ‘effective Hamiltonian’ function to obtain the mean energy equality. We will theoretically show here what kind of equalities would be necessary conditions for self organization of nonextensive systems without using any predefined effective Hamiltonian function. The results will come up as a direct consequence of our approach. Moreover, we derive relations between *q*-renormalized entropy and the generalized relative entropies from the viewpoint of information theoretic approaches in Section 2.

In Section 4, we use a paradigmatic toy model, the logistic map, in order to numerically show the level of self-organization (or the degree of complexity from self-organisation) for a system that paves its way to successive stable branches, and then to successive chaotic band mergings as the parameter of the system slightly changes. We show the behaviour of the *q*-relative entropies in both Bregmann and Csiszar forms as the suitable measure for self-organisation, and define their types of complexity (type-I, -II or -III). Finally, we show a unique $q^{*}$ values obtained through evolution of the states with the system parameter and relate its value at the edge of chaos with the $q_{sensitivity}$ value obtained for the logistic map [28,29].

## 2. Thermodynamic Perspective of the Renormalized Entropy and Connections

Let us consider an evolution from an equilibrium state $p_{0}$ to an arbitrary one *p* as the control parameter of the complex system slightly changes from $a_{0}$ to $a_{0}+\Delta a$. Entropy produced by the change of state (i.e., corresponding information gain) in such an evolution can be given by the Kullback-Leibler relative entropy [30]
(11)$$K(p||p_{0})=\sum_{i}p_{i}\ln\frac{p_{i}}{p_{0i}}\geq 0\;.$$

Adding and subtracting $\sum_{i}p_{0i}\ln p_{0i}$ to and from right-hand side of Equation (Equation 11), this can be rewritten as
(12)$$K(p||p_{0})=-{\Delta S}^{\left(p_{0}\to p\right)}-\sum_{i}\left(p_{i}-p_{0i}\right)\ln p_{0i}$$
where ${\Delta S}^{\left(p_{0}\to p\right)}=\sum_{i}p_{0i}\ln p_{0i}-\sum_{i}p_{i}\ln p_{i}$.

Substituting $p_{0i}=p_{i}^{eq}$, the canonical equilibrium distribution of exponential form in Equation (Equation 2), into the logarithmic function $\ln p_{0i}$ in Equation (Equation 12), it can be immediately shown that
(13)$$K(p||p^{eq})=-{\Delta S}^{\left(p^{eq}\to p\right)}+\beta \Delta {\langle E\rangle }^{\left(p^{eq}\to p\right)}$$
where $\Delta {\langle E\rangle }^{\left(p^{eq}\to p\right)}={\langle E\rangle }^{p}-{\langle E\rangle }^{p^{eq}}$ is the difference between the mean energies through the evolution from the state $p^{eq}$ to the state *p*. Comparing Equations (Equation 12) and (Equation 13), it follows that
(14)$$\Delta {\langle E\rangle }^{\left(p^{eq}\to p\right)}=-\frac{1}{\beta }\sum_{i}\left(p_{i}-p_{i}^{eq}\right)\ln p_{i}^{eq}\;.$$

One can notice that Equations (Equation 12)–(Equation 14) lead three important connections regarding the proofs of Equations (Equation 3)–(Equation 5):(i)Equation (Equation 3), the second law of thermodynamics ${\Delta S}^{\left(p\to p^{eq}\right)}\geq 0$, can be derived from Equation (Equation 13) noticing that ${\Delta S}^{\left(p\to p^{eq}\right)}=-{\Delta S}^{\left(p^{eq}\to p\right)}$ and $K(p||p^{eq})\geq 0$. This derivation requires the limitation that the Gibbs equality holds, i.e., $\sum_{i}\left(p_{i}-p_{i}^{eq}\right)\ln p_{i}^{eq}\propto \Delta \langle E\rangle =0$, which implies the evolution is isolated, i.e., the mean energy is the same through the evolution.(ii)Equation (Equation 4) can easily obtain from Equation (Equation 13) using the definition of the free energy given as $F={\langle E\rangle }^{r}-\frac{1}{\beta }S$ for any state (*r*). It means that Kullback-Leibler relative entropy is associated with the free energy difference of the states in such an evolution.(iii)Equation (Equation 5), the result of the S-theorem by Klimontovich, can be shown from Equation (Equation 13) by a transformation $p^{eq}\to {\tilde{p}}^{eq}$ ensuring the renormalization of the state so that it compensates the mean energy difference between the states corresponding to the renormalized equilibrium and non-equilibrium stationary states, i.e $\Delta {\langle E\rangle }^{\left({\tilde{p}}^{eq}\to p\right)}=0$. The compensation requires the Gibbs equality defined as
(15)$$\sum_{i}p_{i}\ln p_{i}^{eq}=\sum_{i}{\tilde{p}}_{i}^{eq}\ln p_{i}^{eq}.$$

Such a renormalization enables us to use the Gibbs theorem for an open system with energy flux. To compare the states in terms of the renormalized $\Delta \tilde{S}$ and Kullback-Leibler relative entropies *K*, the connection can be written as
(16)$$\Delta \tilde{S}=-K(p||{\tilde{p}}^{eq})\leq 0$$
by means of Equations (Equation 12) and (Equation 15).

It should be noted that our assumption reveals with Equations (Equation 12)–(Equation 14) why a quantity called the effective Hamiltonian, $H_{eff}=-\ln p_{0i}$, for the reference equilibrium state was preferred by Klimontovich. In our assumption, choosing the reference state as $p_{0}=p^{eq}$ yields $\Delta {\langle E\rangle }^{\left(p^{eq}\to p\right)}$ in Equation (Equation 13), spontaneously. The details and applications on both synthetical and real data of the renormalized entropy by Klimontovich can be found in references [10,12,21,31] and [32,33,34,35], respectively.

## 3. Derivation of the *q*-Renormalized Entropy and Connections

Similar to the connection in Equation (Equation 4) between Kullback-Leibler relative entropy and free energy differences, there are two types of *q*-generalized relative entropies whose connections to *q*-free energy differences exists [36]. Respectively, they are called Bregmann form given by
(17)$$\begin{array}{cc} K_{q}^{B}\left(p|\right|p_{0})= & \frac{1}{q-1}\sum_{i}p_{i}\left({p_{i}}^{q-1}-{p_{0i}}^{q-1}\right)-\\ & \sum_{i}\left(p_{i}-p_{0i}\right){p_{0i}}^{q-1} \end{array}$$
and Csiszar form defined as
(18)$$K_{q}^{C}\left(p|\right|p_{0})=\frac{1}{q-1}\sum_{i}p_{i}\left[{\left(\frac{p_{i}}{p_{0i}}\right)}^{q-1}-1\right]\;.$$

Noticing the dependence of the generalized relative entropies on the constraints from Equation (10), we derive the *q*-renormalized entropies for the evolution from a stationary state in the functional forms of the inverse power law (i.e., *q*-exponentials in Equations (Equation 8) and (Equation 9)) within a thermodynamic perspective similar to the Section 2. The crucial point of such an approach is that a predefined effective hamiltonian function is not necessary. Moreover, we show the necessary conditions of self organization in nonextensive open system using a *q*-Gibbsian equality in the following subsections.

### 3.1. Derivation and Connection I: q-Renormalized Entropy and Bregman Form of Relative Entropy

Firstly, we reorganize the Bregman form of the generalized relative entropy in Equation (Equation 17) as
(19)$$\begin{array}{cc} K_{q}^{B}\left(p|\right|p_{0})= & \sum_{i}p_{0i}\ln_{2-q}p_{0i}+\sum_{i}p_{i}\ln_{2-q}p_{i}-\\ \; & q\sum_{i}\left(p_{i}-p_{0i}\right)\ln_{2-q}p_{0i} \end{array}$$
where the identical relation of (2−q)-deformed logarithm, $\ln_{2-q}x=\left(x^{q-1}-1\right)/\left(q-1\right)$, has been used. It should be noted that the same relation leads to the *q*-logarithmic form of the Tsallis entropy in Equation (Equation 7), that is given by
(20)$$S_{q}=-\sum_{i}p_{i}\ln_{2-q}p_{i}\;.$$

Putting this form in Equation (Equation 19), we have a similar expression to Equation (Equation 12), which reads
(21)$$K_{q}^{B}\left(p|\right|p_{0})=-\Delta {S_{q}}^{\left(p_{0}\to p\right)}-q\sum_{i}\left(p_{i}-p_{0i}\right)\ln_{2-q}p_{0i}$$
where $\Delta {S_{q}}^{\left(p_{0}\to p\right)}=S_{q}\left(p\right)-S_{q}\left(p_{0}\right)$ is the change in the Tsallis entropies through an evolution from the state $p_{0}$ to *p*.

Substituting $p_{0i}=p_{i}^{ord}$, the stationary distribution of the (2−q)-exponential form in Equation (Equation 8), into the (2−q)-logarithmic function $\ln_{2-q}p_{0i}$ in Equation (Equation 21), we can immediately write
(22)$$K_{q}^{B}\left(p|\right|p^{ord})=-{\Delta S_{q}}^{\left(p^{ord}\to p\right)}+\beta^{{}^{\prime }}\sum_{i}\left(p_{i}-p_{i}^{ord}\right)\epsilon_{i}$$
where $\beta^{{}^{\prime }}=\frac{q\beta^{*}}{Z^{q-1}}$. From the transformation $p_{i}^{ord}\to {\tilde{p}}_{i}^{ord}$ on the reference state, one can easily obtain
(23)$$K_{q}^{B}\left(p|\right|{\tilde{p}}^{ord})=-{\Delta S_{q}}^{\left({\tilde{p}}^{ord}\to p\right)}+\beta^{{}^{\prime }}\Delta {\langle E\rangle }^{\left(p^{ord}\to p\right)}$$
where $\Delta {\langle E\rangle }^{\left(p^{ord}\to p\right)}={\langle E\rangle }^{p}-{\langle E\rangle }_{q}^{p^{ord}}$ is the mean energy difference and ${\tilde{p}}_{i}^{ord}={\left(p_{i}^{ord}\right)}^{q}/\sum_{i}{\left(p_{i}^{ord}\right)}^{q}$ is the distribution chosen so that it enables us to vanish the second terms in the right hand side of Equations (Equation 22) and (Equation 23) at a unique value of $q=q^{*}$, namely,
(24)$$K_{q^{*}}^{B}\left(p|\right|{\tilde{p}}^{ord})=-{\Delta S_{q^{*}}}^{\left({\tilde{p}}^{ord}\to p\right)}$$

It should be noted here that the transformation enables us to equate the mean energies, i.e., ${{\langle E\rangle }_{q^{*}}}^{p^{ord}}={\langle E\rangle }^{p}$, at a unique value of *q* taking the normalized *q*-average instead of the ordinary average in the calculation of the mean energy of the reference state. In other words, it re-normalizes the mean energy of the reference state.

Comparing Equations (Equation 21)–(Equation 24), it can be easily shown that the compensation requires a *q*-Gibbsian equality given by
(25)$$\sum_{i}p_{i}\ln_{2-q^{*}}p_{i}^{ord}=\sum_{i}{\tilde{p}}_{i}^{ord}\ln_{2-q^{*}}p_{i}^{ord}$$
where the unique $q^{*}$ can be found numerically.

One can easily show that Bregman form of the generalized relative entropy in Equation (Equation 22) is associated with the *q*-generalized version of free energy differences as can be given in Equation ([Disp-formula FD10a-entropy-25-00517]) where $F_{q}={<E>}^{p}-S_{q}/\beta^{{}^{\prime }}$ and $F_{q}^{ord}={<E>}^{p^{ord}}-S_{q}/\beta^{{}^{\prime }}$. Moreover, as can be seen in Equation (Equation 24), there is a one-to-one correspondence between the generalized relative entropy and the *q*-renormalized entropy due to the compensation of the mean energy differences.

It is also worth noting that *q*-renormalized entropy is not a generalization of the usual renormalized entropy by Klimontovich. Although the generalized relative entropy in Equation (Equation 17) recovers the relative entropy by Kullback-Leibler in the limit q→1, the Gibbsian equality in Equation (Equation 25) is ensured at $q^{*}=1$ only if the transition from $p^{ord}$ to *p* belongs a cyclic process or the states possess the same degree of complexity. At $q=q^{*}\neq 1$, the value of *q* holds the Gibbsian equality as the necessary condition of self organization for the transition between distinct states and leads the connection in Equation (Equation 24). Therefore, the parameter $q^{*}$, which is the unique value of *q*, measures the relative degree of order/disorder between the states. We confirm it using the toy model in Section 4.

### 3.2. Derivation and Connection II: q-Renormalized Entropy and Csiszar Type of Relative Entropy

Substituting $p_{0i}=p_{i}^{nor}$, the stationary distribution of the *q*-exponential form in Equation (Equation 9), into Equation (Equation 18) and using the identical relation $Z^{1-q}=C_{q}$ where $C_{q}=\sum_{i}{\left(p_{i}^{nor}\right)}^{q}$, the Csiszar form of the generalized relative entropy can be written as
(26)$$\begin{array}{cc} K_{q}^{C}\left(p|\right|p^{nor})= & -{\Delta S_{q}}^{\left(p^{nor}\to p\right)}\\ & +\hat{\beta }D_{q}\sum_{i}\frac{p^{q}}{D_{q}}\left(\epsilon_{i}-{\langle E\rangle }_{q}^{p^{nor}}\right) \end{array}$$
where $D_{q}=\sum_{i}p_{i}^{q}$. Using the *q*-deformed logarithm form, $\ln_{q}x$, instead of the second term in the right hand side of Equation (Equation 26), it follows that
(27)$$\begin{array}{cc} K_{q}^{C}\left(p|\right|p^{nor})= & -{\Delta S_{q}}^{\left(p^{nor}\to p\right)}\\ & -C_{q}D_{q}\sum_{i}\left(\frac{p^{q}}{D_{q}}-\frac{{\left(p_{i}^{nor}\right)}^{q}}{C_{q}}\right)\ln_{q}\left(p_{i}^{nor}\right) \end{array}$$

By the transformation $p_{i}\to {\tilde{p}}_{i}$ on the other state, Equation (Equation 26) yields
(28)$$K_{q^{*}}^{C}(\tilde{p}\left|\right|p^{nor})=-{\Delta S_{q^{*}}}^{\left(p^{nor}\to \tilde{p}\right)}$$
where ${\tilde{p}}_{i}=p_{i}^{1/q}/\sum_{i}p_{i}^{1/q}$ is the distribution chosen so that it enables us to vanish the second terms in the right hand side of Equations (Equation 26) and (Equation 27) at a unique value of $q=q^{*}$, satisfying ${{\langle E\rangle }_{q^{*}}}^{p^{nor}}={\langle E\rangle }^{p}$.

Comparing Equations (Equation 26) and (Equation 27), it can be easily shown that the compensation of mean energy difference requires a Gibbsian equality given by
(29)$$\sum_{i}p_{i}\ln_{q^{*}}p_{i}^{nor}=\sum_{i}\frac{{\left(p_{i}^{nor}\right)}^{q^{*}}}{C_{q}}\ln_{q^{*}}p_{i}^{nor}$$
where the unique $q^{*}$ can be found numerically.

At this point, it should be emphasized that the Gibbsian equalities in Equations (Equation 25) and (Equation 29) both lead the same mean energy equality, that is ${{\langle E\rangle }_{q^{*}}}^{p_{0}}={\langle E\rangle }^{p}$, if one applies a transformation on the reference state for the Bregman form of the generalized relative entropy taking $p_{0}=p^{ord}$ and on the other state for the Csiszar form of the generalized relative entropy taking $p_{0}=p^{nor}$.

## 4. Application: Logistic Map

To identify behaviour of the *q*-renormalized entropies for an evolution in the control parameter space and to illustrate the consistency of Bregman and Csiszar forms of the relative entropies within the context of the self-organization, we apply these procedures on the logistic map. In addition to its ‘very simple expression’ as a toy model, the logistic map has ‘highly complicated’ dynamics in the phase space [37]. Moreover, it is very convenient to search whether there exist a connection between self-organization and bifurcation processes as the system parameter slightly changes.

The expression of the logistic map reads
(30)$$f\left(X_{t}\right)=X_{t+1}=1-aX_{t}^{2}$$
where $X_{t}\in \left[-1,1\right]$ is a sufficiently long phase space trajectory, *t* is iteration step (t=1,2,…,N) and a∈[0,2] is the control parameter of the map.

For the evolution of the map from $a_{0}$ to $a_{0}+\Delta a$, one can easily generate the trajectories $\left\{X_{t}\left(a_{0}\right)\right\}$ and $\left\{X_{t}\left(a_{0}+\Delta a\right)\right\}$. Respectively, the corresponding probability distributions estimated from the trajectories are $p_{0}=p_{0}\left(X,a_{0}\right)$ for the reference state and $p_{1}=p_{1}\left(X,a_{0}+\Delta a\right)$ for the other state where $\sum p_{0}=\sum p_{1}=1$.

For the estimation of the distributions, we use the dependence of spectral intensities on the frequency *w*. In other words, we use the Fourier transformation $p\left(w,a\right)=F\left(w,a\right)\cdot F^{*}\left(w,a\right)$ of the trajectory $\left\{X_{t}\left(a\right)\right\}$ instead of residence time distribution. Technically, we generate trajectories of the map in Equation (Equation 30) with the length of 65,536 points after 4096 points are discarded as transients. The spectrum is then averaged over 16 periodograms with a length of 4096 points. The details of the estimation procedure can be found in a recent paper [10].

It is well known that the bifurcation diagram of the logistic map represents a very rich dynamics where transitions with period-doubling route to chaos arise as the control parameter changes in the range of a∈[0,2] as can be seen in Figure 2. The map has a critical point at $a=a_{c}=1.401155\ldots$ which can be approached from the most ordered state where the value of the control parameter is a=0. From a=0 (where period-1 occurs) to $a=a_{c}$ (where $2^{\infty }$ periods accumulate), the map shows a period-doubling procedure of $2^{n}$ periods. One can also approach the critical point from the most chaotic state (where the value of the control parameter is a=2), via a band splitting procedure where $2^{\infty }$ bands split at the critical point. In other words, the map is in a periodic region from a=0 up to $a=a_{c}$ with a period doubling procedure as it is in a chaotic region from a=2 up to $a=a_{c}$ with a band splitting procedure. It is also possible to see narrow periodic windows in the chaotic region that possess similar structures to those of the map in the whole range of the control parameter, i.e., a∈[0,2]. Moreover, the Lyapunov exponent of the map can be calculated using (31)$$\lambda =\lim_{N\to \infty }\frac{1}{N}\sum_{t=0}^{N-1}\log\left|f^{\prime }\left(X_{t}\right)\right|\;,$$ by substituting the first derivative of the map function in the Equation (Equation 31) and is used to make a distinction between periodic regions (λ<0) and chaotic ones (λ>0) [38].

To compare the reference state of the map with all other states within *q*-renormalized entropies, we choose the reference state $p_{0}$ at $a=a_{0}=0$ and all other states in the region of a∈[0,2], i.e., $p_{0}=p\left(w,a_{0}=0\right)$ and $p_{1}=p\left(w,a_{0}+\Delta a\right)$ with a parameter increase step Δa=0.01. To calculate the entropies, we firstly numerically define the unique $q^{*}$ values that hold the Gibsian equalities in Equations (Equation 25) and (Equation 29). We then obtain the *q*-renormalized entropies $\Delta S_{q^{*}}^{\left({\tilde{p}}_{0}\to p\right)}$ and $\Delta S_{q^{*}}^{\left(p_{0}\to \tilde{p}\right)}$ that are associated with *q*-relative entropies $K_{q^{*}}^{B}\left(p|\right|{\tilde{p}}_{0})$ and $K_{q^{*}}^{C}(\tilde{p}\left|\right|p_{0})$, respectively.

In Figure 3, from top to bottom, we plot the bifurcation diagram, the Lyapunov exponent, the *q*-renormalized entropy in Bregman and Csiszar forms and evolution of the $q^{*}$ values in the control parameter space. We denote some points just above the bifurcation diagram as can be seen as $a_{0},a_{1},a_{2},\ldots ,{\tilde{a}}_{2},{\tilde{a}}_{1},{\tilde{a}}_{0}$ to divide the map in distinct regions as analogous to the periodic and chaotic band regions in Figure 2. There are $2^{n-1}$ number of periods and chaotic bands between the regions $a\in [a_{n-1},a_{n}]$ and $a\in [{\tilde{a}}_{n-1},{\tilde{a}}_{n}]$ where n=1,2,3,…,∞, respectively. As the control parameter evolves from $a_{0}=0$ to ${\tilde{a}}_{0}=2$, the periodic trajectories bifurcate at the critical point $a_{n}$, where the first bifurcation point is $a_{1}$, up to the chaos threshold $a_{c}$, where infinite number of periods exists. As a reverse process, an infinite number of chaotic bands which exist at the chaos threshold $a_{c}$ start to merge through the critical points ${\tilde{a}}_{n}$ up to the ${\tilde{a}}_{1}$ where the last chaotic band merging exists. The Lyapunov exponent vanishes at all critical points from $a_{1}$ to ${\tilde{a}}_{1}$ as it has a negative value in the range of $a\in [a_{0},a_{c})$ and has a positive value in the range of $a\in (a_{c},{\tilde{a}}_{0}]$. It can also be seen that the *q*-renormalized entropy in both Bregman and Csiszar forms point out the same relative degree of order/disorder in the range of period–1 which implies a low level of self-organization/complexity. When the sequence of branches emerges at $a_{1}$, the relative order, i.e., the level of self-organization/complexity, increases and the *q*-renormalized entropies decrease through successive bifurcations up to the chaos threshold point $a_{c}$. Such behaviour of *q*-renormalized entropy in the control parameter space of $a\in [a_{1},a_{c}]$ is compatible with that of the entropy-based measures of type-III in Figure 1. In the range of chaotic band merging area of $a\in (a_{c},{\tilde{a}}_{1}]$ as a reverse process, increase in *q*-renormalized entropies corresponds to the entropy-based measures of type-I in Figure 1. It means that the relative order, i.e., the level of self-organization/complexity, decreases through the band merging area. It should be noted that the *q*-renormalized relative entropy in Csiszar form evaluates that the level of order in the range of period–1 of $a\in [a_{0},a_{1}]$ has the same degree of complexity with the level of disorder in the range of chaotic band–1 of $a\in ({\tilde{a}}_{1},{\tilde{a}}_{0}]$. However, the degree of order/disorder in the range of chaotic band–1 of $a\in ({\tilde{a}}_{1},{\tilde{a}}_{0}]$ decreases/increases except for a very thin periodic window, which is similar to the behaviour of the Lyapunov exponent in the same range of the control parameter. Moreover, *q*-renormalized entropy in Bregman form is more accurate for localization of the chaos threshold such that it corresponds to a local minimum between $a\in (a_{1},{\tilde{a}}_{1}]$ where $a_{c}=1.4011\ldots$.

The equalities between the *q*-generalized relative entropies and the *q*-renormalized entropies in Equations (Equation 24) and (Equation 28) guarantee that the evolution of the *q*-generalized relative entropies as complexity measures in the range of period doublings and chaotic band mergings of $a\in [a_{1},{\tilde{a}}_{1}]$ at a unique $q=q^{*}$ value conforms with the behaviour of the entropy-based measure of type-II in Figure 1. Such behaviour of the complexity function is similar to the behaviour of complexity in coffee automaton (or to experiment of coffee with milk). It was discussed by defining a “complextropy” measure that first increases and then decreases in closed thermodynamic systems, in contrast to usual Shannon entropy (which increases monotonically) [39]. Similar to the model, in the range of $a_{1}<a_{c}<{\tilde{a}}_{1}$, the $q^{*}$-generalized relative entropy represents the most organized spatial pattern at the chaos threshold where $a=a_{c}$ due to the relations roughly $K_{q^{*}}=-\Delta {\tilde{S}}_{q^{*}}$ where $\Delta {\tilde{S}}_{q^{*}}$ is the general definition of the *q*-renormalized entropy. *q*-relative entropy is an evaluation of the change in entropy relative to a reference state chosen. For the transition between the reference equilibrium state and the other arbitrary state, one can numerically localize the unique $q^{*}$ value as the one for which the Gibbsian equalities hold. We show in the bottom of Figure 3 that the $q^{*}$ values are the same for both Bregman and Csiszar forms of the *q*-renormalized entropies for an evolution in the control parameter space of the logistic map, satisfying the Gibbsian equalities as the necessary condition of self organization. In other words, renormalization enables us to equate mean energies of the states at a unique $q^{*}$ value in a manner that the evolution of the reference state is isolated after renormalization. For the evolution of the $q^{*}$ values in the range of the control parameter a∈[0,2], the $q^{*}$ values decrease from $q^{*}=1$ to $q^{*}=0$ where the maximum value indicates the most ordered state (period-1) and the minimum value points out the most disordered (strongly chaotic) state. The process offers a method by means of *q*-renormalized entropies on how to measure the level of self organization in spatially-extended fractals. On the other hand, the Shannon entropy leads to an increase since it is proportional to the logarithm of the accessible volume in phase space, however a decrease in entropy is necessary to link a connection to the self-organization.

In Figure 4, we zoom to the chaos threshold in order to localize unique $q^{*}$ value at the critical point. It is intriguing that this unique $q^{*}$ value happens to coincide with the $q_{sensitivity}$ value [28,29] (i.e., $q^{*}≃0.2445$). At this point, it is worth noting that this kind of varying *q* parameter tendency with the control parameter of the map is very reminiscent to the behaviour of the running *q* parameter with the energy scale detected in recent cosmological studies [40,41].

## 5. Conclusions

It is well known that the second law of thermodynamic (ΔS≥0) is only valid for an isolated evolution of an arbitrary state to an equilibrium state. This inequality can be derived by substituting Gibbs equality in the definition of Kullback-Leibler relative entropy, which implies that the equilibrium state shows the greatest disorder (or chaoticity) as compared to any arbitrary state as long as the mean energy is the same. The mean energy equality through the evolution is a consequence of Gibbs equality, that points out a strong limitation of the law. Hence, it is violated for the evolution of open systems in which the exchange of energy/matter with its surroundings is allowed. The problem was solved by Klimantovich via the S-theorem where ’S’ stands for criterion of self-organization. The theorem is based on renormalization of the equilibrium distribution in a manner that Gibbs equality holds. Mean energy in terms of a predefined effective Hamiltonian function for an open system is constant through the evolution after renormalization. The renormalization on the distribution leads a renormalized entropy as a new complexity measure to compare distinct states, i.e., a renormalized equilibrium state and an arbitrary one. For an isolated evolution from the renormalized equilibrium state to an arbitrary one, a decrease in the renormalized entropy indicates an increase in the relative degree of order in the system that indicates the creation of complicated structures in the phase space as first suggested by Haken in the context of self-organization [13]. Although the renormalized entropy is a suitable measure to explain highly organised structures that emerge in phase space, we have shown that its expression ($\Delta \tilde{S}=-KL\leq 0$) is valid for the systems which evaluate from canonical equilibrium state. Moreover, choosing a reference state in exponential form spontaneously reveals the predefined effective Hamiltonian function ($H_{eff}=\ln p_{0}$) by Klimontovich directly from the calculations. We have also shown that such kind of relation between *q*-renormalized entropy and *q*-generalized relative entropies (in the form of both Bregmann and Csiszar) can be written by introducing new *q*-Gibbsian equalities as the necessary conditions of self-organisation. The crucial point for the new equalities is that they are only valid for a unique $q=q^{*}$ value for the transition between two states and lead to the same mean energy equality that is ${{\langle E\rangle }_{q^{*}}}^{p_{0}}={\langle E\rangle }^{p}$. To achieve this result, it is necessary to apply a transformation on the reference state for the Bregman form of the generalized relative entropy taking $p_{0}=p^{ord}$ and on the other state for the Csiszar form of the generalized relative entropy taking $p_{0}=p^{nor}$ as the stationary distributions of the (2−q)-exponential and *q*-exponential forms, respectively. To verify the results numerically, we have used the control parameter evolution of the logistic map. As the control parameter changes in a∈[0,2], we have shown a fall in the *q*-renormalized entropies through period doublings in the range of $a\in [0,a_{c}]$ and an increase in the *q*-renormalized entropies through chaotic band mergings in the range of $a\in [a_{c},2]$. Such kind of behaviour of the *q*-renormalized entropy is compatible with the SDL complexity of type-III and type-I as the signs of emerging and destroying highly organized structures in the phase space, respectively [5]. We have also looked closely at the chaos threshold of the map, and interestingly we discovered that the unique $q^{*}$ value is $q^{*}≃0.2445$, which coincides with the value of $q_{sensitivity}$ given in the literature [28,29].

Finally, it would be good to note that these considerations could be applied to some specific class of nonextensive systems, such as black holes and other gravitational systems. An interesting future work addressing a possible discussion of our scheme for such systems would be highly welcomed.

## Figures and Tables

**Figure 1 entropy-25-00517-f001:**
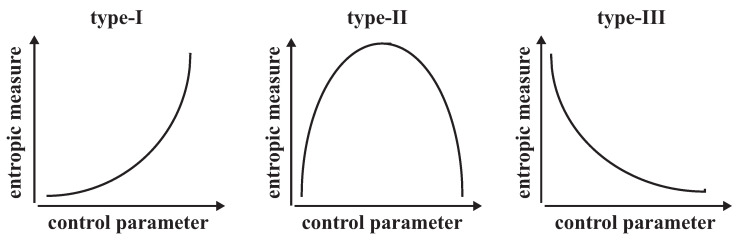
An example of types of entropy based measures as a function of control parameter (adapted from [5]).

**Figure 2 entropy-25-00517-f002:**
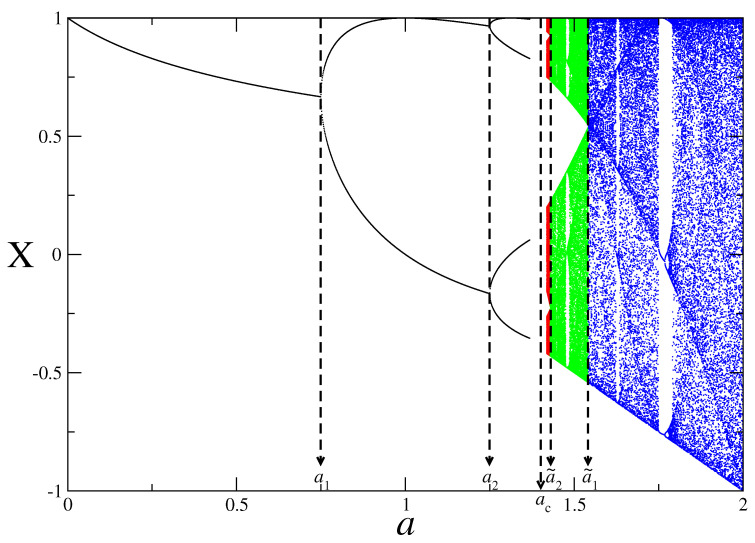
A representation of the pitchfork bifurcations in periodic regime (black) and the band merging structures in chaotic regime (blue, green and red) of the logistic map. The black dashed lines represent the bifurcation points ($a_{n}$) and the band merging points (${\tilde{a}}_{n}$).

**Figure 3 entropy-25-00517-f003:**
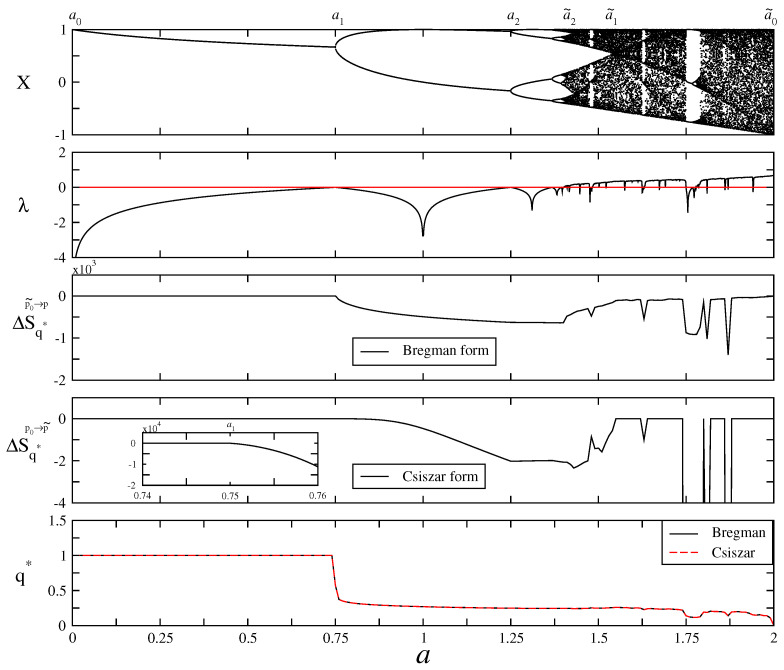
For the evolution of the logistic map in the control parameter space, from top to bottom: Bifurcation diagram, Lyapunov exponent, Bregman form of *q*-renormalized entropy, Csiszar form of *q*-renormalized entropy and $q^{*}$ values that hold the Gibbsian equalities in Equations (Equation 25) and (Equation 29), respectively.

**Figure 4 entropy-25-00517-f004:**
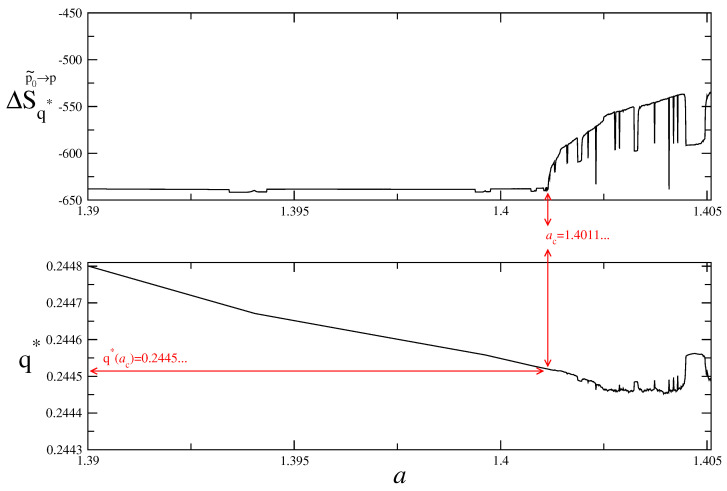
*q*-Renormalized entropy (top) and $q^{*}$ values for the control parameter evolution of the logistic map near chaos threshold (a∈[1.390,1.4051]).

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
