# Peer review of "Necessary Condition of Self-Organisation in Nonextensive Open Systems"

_entropy, 2023, doi:10.3390/e25030517_

Round 1
Reviewer 1 Report
In this manuscript titled “Necessary condition of self-organization in nonextensive open systems” the authors analyze the evolution of nonextensive open systems from an equilibrium state to an arbitrary one by using q-exponential distribution. In particular, they introduce new q-Gibbsian equalities as necessary condition of self-organization in these systems and comment on the derivation of the connections between q-renormalized q-relative entropies and in both Bregman and Csiszar forms. To numerically verify their results, they finally use the logistic map and connect the map parameter to nonextensive one.
In my opinion the manuscript is well-written and self consistent. It sounds mathematically rigorous as I checked some computations and they seem to be correct. Also, the work provides a non-trivial contribute toward understanding non-equilibrium physics of nonextensive open systems. Before definitely recommending for publication, however I have the two following suggestions:
1) To improve application section and give broader view to their study, the authors could apply their considerations to some specific class of non-extensive systems. For instance, black holes and in general gravitational systems are expected to obey non-extensive statistics. It is interesting to discuss the present results in that context.
2) Although not contemplated in the original formalism by Tsallis, some recent works have argued that in renormalized quantum field theory and cosmology non-extensive parameter may be running with the energy scale, see
Eur. Phys. J. C 79, 242 (2019)
Phys. Rev. D 104, 045004 (2021)
The authors could mention these works and comment on how/whether their conclusions are generalized in the presence of a varying q-exponent.
I will be able to reconsider the manuscript for publication after the above suggestions have been properly taken into account.
Author Response
RESPONSES TO THE FIRST REVIEWER'S COMMENTS
"In my opinion the manuscript is well-written and self consistent. It sounds mathematically rigorous as I checked some computations and they seem to be correct. Also, the work provides a non-trivial contribute toward understanding non-equilibrium physics of nonextensive open systems."
--> We thank to the Referee for his/her critical reading of the manuscript. We appreciate his/her comments and included them in the new version of the manuscript.
"1) To improve application section and give broader view to their study, the authors could apply their considerations to some specific class of non-extensive systems. For instance, black holes and in general gravitational systems are expected to obey non-extensive statistics. It is interesting to discuss the present results in that context."
--> We have added a sentence about this in page 13, lines 396–399.
"2) Although not contemplated in the original formalism by Tsallis, some recent works have argued that in renormalized quantum field theory and cosmology non-extensive parameter may be running with the energy scale, see Eur. Phys. J. C 79, 242 (2019) Phys. Rev. D 104, 045004 (2021). The authors could mention these works and comment on how/whether their conclusions are generalized in the presence of a varying q-exponent."
--> We have added a sentence and appropriate references ([30,31]) in page 11, lines 350–353.
Reviewer 2 Report
The paper is a good introduction and explanation into the field of
self-organization.
The developments of the ongoing formulae are correct; I could not find any
error in the text.
The 'toy' example of the well-known logistic map is good understandable,
and it is a good case for a comparison of the used theoretical tools.
Therefore, I judged that the manuscript is suitable for publication. When the
authors are planning to improve this manuscript further for submission,
I would be grateful if the authors would consider the following points.
page 4, line 110, I do not understand the right constraint.
page 5, line 139, what means 'in as' ?
page 9, line 285, the calculation of the 'Lyapunov exponents' should be
explained, or at least a reference should be given.
Author Response
RESPONSES TO THE SECOND REVIEWER'S COMMENTS
"The paper is a good introduction and explanation into the field of self-organization. The developments of the ongoing formulae are correct; I could not find any error in the text. The “toy” example of the well-known logistic map is good understandable, and it is a good case for a comparison of the used theoretical tools. Therefore, I judged that the manuscript is suitable for publication. When the authors are planning to improve this manuscript further for submission, I would be grateful if the authors would consider the
following points."
--> We are very grateful to the Referee for the careful and critical reading of the manuscript. We appreciate these comments and the positive assessment about the manuscript. We have considered all recommendations of the Referee and all changes are written with red text color in the present version of
the manuscript.
"1) page 4, line 110, I do not understand the right constraint."
--> For clarity, we have added a sentence and a reference ([25]) in page 4, lines 118–120.
"2) page 5, line 139, what means “in as” ?"
--> We have deleted the “in” which was in fact a typo in the text.
"3) page 9, line 285, the calculation of the “Lyapunov exponents” should be explained, or at least a reference should be given."
--> We have added definition of the Lyapunov exponent (Eq. 31) and a reference ([40]) in page 9, lines 279–282.
Round 2
Reviewer 1 Report
The new version meets criteria to be published in this journal.